# Integration Analysis of circRNA–miRNA–mRNA and Identification of Critical Networks in Valgus-Varus Deformity (*Gallus gallus*)

**DOI:** 10.3390/genes14030622

**Published:** 2023-03-01

**Authors:** Jianzeng Li, Yanchao Ma, Chunxia Cai, Lujie Zhang, Xinxin Liu, Ruirui Jiang, Donghua Li, Zhuanjian Li, Xiangtao Kang, Yadong Tian, Ruili Han

**Affiliations:** 1College of Animal Science and Technology, Henan Agricultural University, Zhengzhou 450046, China; 2Key Research Project of the Shennong Laboratory, Zhengzhou 450002, China

**Keywords:** valgus-varus deformity, RNA-seq, broiler, competing endogenous RNA, skeletal development

## Abstract

Valgus-valgus deformity (VVD) is a common leg deformity in broilers with inward or outward deviation of the tibiotarsus and tarsometatarsus. The competing endogenous RNA (ceRNA) network plays an essential role in the study of leg disease. However, its role in the etiology and pathogenesis of VVD remains unclear. Here, based on case (VVD) and control (normal) group design, we performed analyses of differentially expressed circRNAs (DEcircRNAs), differentially expressed miRNAs (DEmiRNAs) and differentially expressed mRNAs (DEmRNAs). Transcriptome data derived 86 DEcircRNAs, 13 DEmiRNAs and 410 DEmRNAs. Functional analysis showed that DEmRNAs were significantly enriched in cell cycle, apoptosis, ECM-receptor interaction, FoxO signaling pathway and protein processing synthesis. DEcirc/miRNA-associated DEmRNAs were associated with skeletal and muscle growth and development pathways, including mTOR, Wnt, and VEGF signaling pathways. Subsequently, a circRNA–miRNA–mRNA regulatory network was constructed based on the ceRNA hypothesis, including 8 circRNAs, 6 miRNAs, and 31 mRNAs, which were significantly enriched in the skeletal developmental pathway. Finally, two key mRNAs (*CDC20* and *CTNNB1*) and their regulatory axes were screened by the PPI network and cytohubba. The expression levels of *CDC20* and *CTNNB1* in cartilage and seven other tissues were also quantified by qPCR. In conclusion, we analyzed the functions of DEmRNA, DEcircRNA and DEmiRNA and constructed the hub ceRNA regulatory axis, and obtained two hub genes, *CDC20* and *CTNNB1*. The study more deeply explored the etiology and pathogenesis of VVD and lays the foundation for further study of the role of the ceRNA network on skeletal development.

## 1. Introduction

Valgus-valgus deformity (VVD) is a prevalent long bone deformity in broilers, characterized mainly by valgus (outward angulation) or varus (inward angulation) of the tibiotarsus and tarsometatarsus [1,2,3]. Due to its special posture, it was initially known as “leg twist,” now abbreviated as VVD [4]. Broilers with VVD show symptoms such as lameness, pain, slimness, and inflammation, which seriously affect the welfare of broilers, and increase economic loss [5,6,7]. Early studies have shown that the incidence of VVD is as high as 0.5–2% and increases with breeding time. Although unsuitable light, lack of exercise, uncontrolled feeding, biomechanics, and genetics are some of the factors that contribute to the incidence of VVD [5,6], its exact etiology and pathogenesis remain unclear. In addition, early studies have found that VVD has some heritability and that the incidence of VVD can be reduced by genetic selection. Therefore, starting from genetic factors will be more beneficial to exploring the etiology of VVD.

The term non-coding RNA (ncRNA) is commonly employed for RNA that does not encode a protein, but has potential to convey information and perform functions at the RNA or protein levels [8]. CircularRNA (circRNA) is an emerging ncRNA with a highly stable closed-loop structure formed by reverse splicing at the 3′ and 5′ ends, which is formed mainly by exons, introns or intergenic regions [9,10,11]. MicroRNA (miRNA) is a class of endogenous non-coding RNA, around 21–25 nucleotides in length [12]. The seed region of the miRNA binds to the 3′UTR target of the mRNA, thereby mediating post-transcriptional regulation of the target gene [13]. The competing endogenous RNA (ceRNA) hypothesis, first proposed by Salmena et al. [14], suggests that circRNA serves as a molecular sponge by the common miRNA response element that indirectly regulates downstream target genes [14,15].

Recently, ceRNA (circRNA–miRNA–mRNA) has been reported to play an important role in cell proliferation, cell differentiation, ECM synthesis and degradation, immune response, and growth and development [16,17,18,19,20]. Studies have shown that the circ-0005526/miR-142-5p/TCF4, circ-UBE2G1/miR-373/HIF-1α, circ-0000423/miR-27b-3p/MMP-13, and circ-SEC24A/miR-26b-5p/DNMT3A regulatory axis can promote inflammatory factor-induced chondrocytes apoptosis and extracellular matrix degradation [18,21,22,23]. Then, the circ-LRP1B/miR-34a-5p/NRF1 regulatory axis can promote chondrocytes proliferation and inhibit their apoptosis [24]. Secondly, circ-4099/miR-616-5p/SOX9 and circSNHG5/miR-496-3p/CITED2 can promote the synthesis and secretion of extra chondrocytes matrix in intervertebral disc degenerative lesions [25,26]. However, the circRNA–miRNA–mRNA interaction in the pathogenic mechanisms of VVD is still unknown. 

In this study, we performed and analyzed the circRNAs and miRNAs transcriptome data of Hubbard broilers between VVD and normal broilers, and the DEmRNAs were re-annotated by Ga6 version. A comprehensive analysis of differentially expressed (DE)circRNA, DEmiRNA and DEmRNA functions was performed by Gene Ontology Biological Processes (GO-BP), Kyoto Encyclopedia of Genes and Genomes (KEGG), Gene Set Enrichment Analysis (GSEA) and Protein–Protein Interaction (PPI) protein interactions, and a ceRNA network was constructed to explore the potential regulatory relationships among the three (Figure 1A). This study aims to screen the ceRNA regulatory axis and key genes associated with cartilage development through transcriptomic studies, laying the foundation for an in-depth investigation of the pathogenesis of the VVD broiler.

## 2. Materials and Methods

### 2.1. Animal and Tissue Collection

Six 35-day-old VVD and healthy Hubbard broilers (three VVD and three normal) of the same sex (male) were obtained from the Zhonghong Sanrong Group Co., Ltd., Tangshan, China, and were selected for subsequent experiment by clinical presentation and anatomical characteristics. The broilers were euthanized by blood loss. Cartilage was collected from the VVD and normal broilers, immediately stored temporarily in liquid nitrogen and transferred to −80 °C until RNA extraction.

### 2.2. Total RNA Isolation and Construction of RNA-seq Libraries

Extraction of total RNA from 6 cartilage tissues using the classical TRIzol extraction method, and the construction of 6 RNA libraries, were performed. The concentration and integrity of the total RNA were measured using a NanoPhotometer^®^ spectrophotometer (Thermo Scientific, Shanghai, China) and RNANano6000 Assay Kit from the Agilent Bioanalyzer 2100 system (Agilent, Beijing, China). Qualified RNA samples were used to construct RNA-seq libraries; the rRNA of those samples was removed using the NEB Next^®^ Ultra™ Directional RNA Library Prep Kit for Illumina ^®^ (NEB, Ipswich, MA, USA). PCR enriched cDNA were of approximately 150-200 bp length. Sequencing library quality was assessed using the Agilent Bioanalyzer 2100 system, and libraries meeting the quality criteria were sequenced using an Illumina Hiseq 4000 platform from Novegene Bioinformatics Technology Co., Ltd. (Beijing, China). The Illumina sequencing generated paired-end reads of 150 bp length.

### 2.3. Screening of DEmRNAs, DEmiRNAs, and DEcircRNAs

We re-annotated the cartilage tissue mRNAs with a Ga6 version and used R software and online site to analyze mRNAs, circRNAs and miRNAs. In brief, to avoid high false-positives in circRNA identification [27], circRNAs were identified jointly by find_circ [28] and CIRC2 [29] software. Screening of sRNAs of 15-25 nt length, and comparison with the miRBase database (https://mirbase.org/, accessed on 12 November 2022) was performed to derive miRNAs. The “DESeq2” package was used for differential analysis of the identified circRNAs and miRNAs. Values of *p* value < 0.05 and |log_2_FC| ≥ 1 were defined to collect the DEmRNAs, DEcircRNAs and DEmiRNAs. Volcano plots and cluster plots were drawn using Hiplot (https://hiplot.com.cn, accessed on 12 November 2022) for DEmRNAs, DEcircRNAs and DEmiRNAs differential type and clustering analysis.

### 2.4. DEmRNA Functional Analysis

The “clusterProfiler” package of R (Version: 4.2.2)software was used for Gene Ontology Biological Processes (GO-BP, http://www.geneontology.org/, accessed on 21 November 2022) annotation, and the Kyoto Encyclopedia of Gene and Genomes database (KEGG, http://www.genome.jp/kegg, accessed on 22 November 2022) for pathway enrichment analysis of DEmRNAs. A confidence level of *p* < 0.05 was considered significantly enriched. The top 10 GO and KEGG pathway results were plotted using the “ggplot2” R package. After that, we performed GSEA analysis of mRNAs and selected the top 10 GSEA analysis results for visualization by “enrichplot” R package.

### 2.5. Construction of DEmRNA PPI Network and Modular Analysis

The interaction between DEG-encoded proteins was analyzed by STRING (version 11.5; https://cn.string-db.org/, accessed on 25 November 2022) online database [30]. The confidence level of the Protein–Protein Interaction (PPI) network was set to >0.4. Cytoscape (version 3.9.0) software was used for graphic optimization and presentation, and the hub module genes were analyzed and extracted by MCODE. The score > 4 hub module genes were selected for display and their KEGG pathways were analyzed using the online website (https://hiplot.com.cn, accessed on 25 November 2022).

### 2.6. DEmiRNA-DEmRNA and DEcircRNA-DEmRNA Regulated Relationship

Prediction of DEmiRNA target genes by miRanda, PITA and RNAhybrid databases to screen for possible DEmiRNA–DEmRNA regulatory relationships was performed. The predicted miRNA–gene relationship was integrated with DEmRNA to obtain DEmiRNA–DEmRNA regulatory relationships. Based on the correspondence between DEcircRNAs and the parent genes, the DEcircRNA–DEmRNA relationships were derived, which were integrated with DEmRNAs to obtain DEcircRNA–DEmRNA regulatory relationships.

### 2.7. KEGG Enrichment Analysis of DEcircRNAs, and DEmiRNAs

KEGG functional analysis of DEmiRNA–DEmRNA and DEcircRNA–DEmRNA pairs was performed by “clusterProfiler” package. The pathway results of *p* < 0.05 of miRNA and circRNA were most significantly enriched. The top 10 functional analysis results were visualized using bubble maps.

### 2.8. Construction of the circRNA–miRNA–mRNA Network

MiRanda and TargetFinder databases was used to predict the miRNA binding sites of the sheared circRNAs (score > 140; energy < -10). The predict results and DEmiRNA intersected to obtain overlapping miRNAs. miRanda, PITA and RNAhybrid databases were used to predict the target genes of the overlapping miRNAs; only the mRNAs present in three of the above databases were used as target genes for overlapping miRNAs [31]. The target genes overlapped with DEmRNA. Finally, we constructed the circRNA–miRNA–mRNA network by using the “ggalluvial” package of R (Version: 4.2.2) software.

### 2.9. QPCR Validation of DEcircRNAs and DE miRNAs

To verify the accuracy and reliability of high-throughput sequencing, we performed qPCR validation of the six DEcircRNAs and five DEmiRNAs with the highest degree of dysregulation. All primer sequences are shown in Table 1. For DEcircRNAs, after treatment with RNase R, RNA was reverse-transcribed into cDNA using PrimeScript™ RT Master Mix (Vazyme, Nanjing, China) containing random primers. For DEmiRNAs, reverse transcription of total RNA into cDNA was performed using specific stem-loop sequences. Using the SYBR fluorescent dye method (5 μL of SYBR^®^ Premix Ex TaqⅡ (Vazyme, Nanjing, China); 0.5 μL of each primer; 3 μL RNase-free water; 1 μL cDNA), qPCR was performed under conditions of 95 °C for 5 min, 95 °C for 15 s, 60 °C for 30 s, 72 °C for 15 s, and 72 °C for 5 min, and all reactions were performed in triplicate. Subsequently, Sanger sequencing vrified the junction seq and loop structures of the DEcircRNAs.

### 2.10. Expression profiles of CDC20 and CTNNB1

The total RNA from heart, liver, kidney, duodenum, pectoral muscle, leg muscle and cartilage was extracted using the traditional TRIzol method and reverse transcribed into cDNA using the kit (Vazyme, Nanjing, China). qPCR reaction conditions were set to 40 cycles of 95 °C for 30 s, 95 °C for 10 s, 60 °C for 30 s, 95 °C for 15 s, 60 °C for 1 min and 95 °C for 15 s. The primer sequences are shown in Table 1. Three technical replicates were performed for each sample, and the expression levels of CDC20 and CTNNB1 in the tissues were calculated using the 2-method method.

## 3. Results

### 3.1. Identification of DEcircRNAs and DEmiRNAs in Chicken Leg Cartilage

We obtained 1145 circRNAs, 650 miRNAs and 12,440 mRNAs (annotation via Ga6 version). At a screening condition of *p* value < 0.05 and |log_2_FC| ≥ 1, 86 DEcircRNAs (44 upregulated and 42 downregulated), 13 DEmiRNAs (9 upregulated and 4 downregulated) and 410mRNAs (136 upregulated and 274 downregulated) were identified by DEseq2 analysis. Analysis of DEcircRNAs sources revealed that 77 DEcircRNAs originated from exons, 8 from intergenic regions, and 1 from introns (Figure 1B). Based on the results of differential expression analysis, volcano and clustering plots were constructed using DEcircRNAs, DEmiRNAs and DEmRNA (Figure 1C–H).

### 3.2. Functional Enrichment Analysis of DEmRNAs

We performed GO and KEGG analysis for differential genes and GSEA analysis for all genes. Upregulated genes were enriched in 516 GO-BP terms and 15 KEGG pathways, and downregulated genes were enriched in 555 GO-BP terms and 20 KEGG pathways. In addition, GSEA was enriched to a total of 250 GO terms and 24 KEGG pathways. Figure 2A–D displays only the top 10 GO terms or KEGG pathways enriched by upregulated or downregulated DEmRNAs. From the results, we found that the upregulated genes were mainly involved in cell apoptotic process, ECM–receptor interaction and the FoxO signaling pathway. Additionally, the downregulated genes were significantly associated with cell cycle process, the cell cycle pathway and the steroid biosynthesis pathway. Finally, GSEA results showed that these genes were related to the cell cycle, mitosis, protein processing in endoplasmic reticulum and steroid biosynthesis (Figure 2E,F). In summary, these DEmRNAs were significantly related to the growth, development, and proliferation of chondrocytes.

### 3.3. PPI Network and Module Function Analysis

A PPI network of DEmRNAs was constructed by STRING database and Cytoscape software, which consisted of 370 notes and 2663 edges. The MCODE in Cytoscape software is an important plug-in for obtaining key sub-network modules. Significant sub-networks with strong interactions were obtained from PPI networks using the MCODE plug-in (degree cut-off: 2; node score cut-off: 0.4; K-Core: 2; max. depth: 100), and seven were selected for display (Figure 3). The modules were analyzed for KEGG pathway and the highest scores were significantly enriched to cell cycle, P53 signaling pathway, and cellular senescence. The remaining modules were significantly enriched in glycolysis, carbon metabolism, amino acid synthesis, steroid biosynthesis, RNA transport, and vesicle transport and phagocytosis.

### 3.4. Enrichment Analysis on Overlapping Genes

To explore the functions of circRNAs and miRNAs, we overlapped the parent genes of DEcircRNAs and target genes of DEmiRNAs with DEmRNAs, and found 13 overlapping genes in the DEcircRNA–DEmRNA relationship and 31 overlapping genes in the DEmiRNA–DEmRNA relationship. GO-BP and KEGG results showed that maternal overlapping genes were mainly enriched in vascular endothelial growth factor, protein kinase D signaling, Rap1, mTOR, and regulation of actin cytoskeleton (Figure 4A,B). miRNA overlapping genes were mainly enriched in neural development, plasma lipoprotein particle organization, melanin and steroidogenesis, tyrosine metabolism, mTOR, and Wnt signaling pathways (Figure 4C,D).

### 3.5. Construction of the circRNA–miRNA–mRNA Regulatory Network

The ceRNA network can provide a deeper understanding of the chondrogenic mechanisms of VVD. Given the potential regulatory role of circRNAs in competitively binding miRNAs to regulate the expression of target genes, we predicted 954 miRNA“sponges” of circRNA using Miranda and Target databases. Among those, 32 circRNAs–miRNAs pairs were identified after intersecting with DEmiRNAs, including 27 circRNAs and 9 miRNAs. These 9 miRNAs further targeted 387 mRNA genes by taking the three databases—Miranda, PITA and RNAhybrid—among which, 31 mRNAs were overlapped with the DEmRNAs of annotating via Ga6 versions. Eventually, the Sankey diagram was used to show the correspondence of the three databases (Figure 5A,B). Functional enrichment analysis showed that these genes were mainly related to neural development and bone growth (Figure 5D,E). Following the ceRNA principle, two circRNAs–miRNAs–mRNAs pairs (novel_circ_0002771/gga-let-7g-3p/*CDC20*, and novel_circ_0000396/gga-miR-193a-3p/*CTNNB1*) were screened (Figure 5C), among which *CDC20* were hub genes in DEmRNA, and *CTNNB1* were genes related to bone development.

### 3.6. qPCR Validation

Six significant DEcircRNAs (novel_circ_0001295, novel_circ_0001037, novel_circ_0000465, novel_circ_0004128, novel_circ_0005122, novel_circ_0003743) and five miRNAs (gga-miR-122-5p, gga-let-7g-3p, gga-miR-132b-5p, gga-miR-193a-3p, gga-miR-2188-5p) were selected for validation of the accuracy of the sequencing data. The expression and back-splicing sites of the circRNAs were validated by divergent reverse-transcription PCR, RNaseR digestion and qRT-PCR, according to previously described methodologies. Divergent primers and convergent primers were designed to amplify six DEcircRNAs in cDNA and genomic DNA samples. The convergent primer from each circRNA produced a single band of a specific length in both cDNA and genomic DNA, whereas the divergent primer only produced a single band in cDNA, suggesting the presence of back-splicing junctions (Figure 6A). PCR products of divergent primers were further detected by Sanger sequencing to confirm the back-splicing junctions (Figure 6B). Based on the resistance of circRNA to RNase R digestion, we quantified six DEcircRNAs with RNase R. The qPCR results showed no significant difference between the RNase R-treated group of DEcircRNA and the control group, however, the internal reference genes GAPDH and β-actin were significantly reduced (Figure 6C). This showed that these DEcircRNAs were more resistant to RNase R relative to linear mRNA. In addition, the six DEcircRNAs and five DEmiRNA qPCR identifications were consistent with the TPM standardized sequencing results (Figure 6D).

### 3.7. Tissue Expression Profiles of CDC20 and CTNNB1

CDC20 is a key gene in the cell cycle, and CTNNB1 is an adhesion-related gene in the wnt/β--catenin signaling pathway, both of which play important roles in cell growth and cartilage development. We further explored the expression of two key genes, *CDC20* and *CTNNB1*, in different tissues (heart, liver, spleen, kidney, duodenum, pectoral muscle, hamstrings and cartilage). The results showed that *CDC20* and *CTNNB1* were highly expressed in the spleen, kidney and duodenum, as well as cartilage. Compared with the healthy group, the expression of the *CDC20* gene was significantly lower in the kidney, leg muscles and cartilage in the VVD group (*p* < 0.01), and the expression of *CTNNB1* gene was significantly lower in the spleen of the VVD group (*p* < 0.05) (Figure 7).

## 4. Discussion

Broiler leg disease is one of the important factors affecting broiler welfare and industrial development. VVD is a critical broiler leg disease with complex etiology, however, there are few studies on the molecular mechanism of VVD [6]. Previous studies have shown that heritability of VVD is 0.21-0.40, bone quality ranges from 0.10 to 0.77 [32,33,34], and genetic selection can significantly reduce the incidence of VVD. Therefore, it is worthwhile studying the molecular etiology of VVD by transcriptomics.

In this study, we found 410 DEmRNAs (Ga6), 85 DEcircRNAs and 13 DEmiRNAs. Direct enrichment functional analysis showed that the upregulated genes were mainly related to chondrocyte proliferation, apoptosis, and extracellular matrix formation, while the downregulated genes were mainly related to chondrocyte cycle and chondrocyte maturation-related amino acid and protein processing and synthesis. Functional analysis of the highest fraction of hub modules identified by PPI protein interaction analysis was performed, equally enriched to cell cycle and P53 signaling pathways. miRNA and circRNA-related gene enrichment analysis in vascular endothelial growth factor revealed mTOR and Wnt signaling pathways. Earlier studies showed diminished cell proliferation in methylprednisolone-induced necrotic cartilage tissue of broiler femoral heads [35]. Fomesin-induced 6 day transcriptome analysis of tibial chondrodysplasia showed that differential genes were mainly enriched in cell proliferation and protein hydrolysis [36]. We speculated that cartilage tissue in the VVD group showed abnormal proliferation and diminished chondrogenic protein synthesis.

Circular RNA (circRNA) is a covalently closed endogenous biomolecule in eukaryotes that typically functions as a molecular sponge for miRNAs with widespread distribution and various cellular functions, which, in turn, affects downstream gene and protein expression patterns [14,15]. Studies show that circPVT1/circ_0058792 acts on miR-21-5p/miR-181a-5p, and thereby mediates Smad7 expression to alleviate steroid-induced femoral head necrosis and regulate its osteogenic differentiation [16,17]. Circ_0000423/miRNA-27b-3p/MMP-13 axis regulates cartilage ECM synthesis in osteoarthritis [18], and exosome-transported circRNA_0001236 enhances chondrogenesis and inhibits cartilage degeneration via miR-3677-3p/Sox9 axis [19]. Here, eight DEcircRNAs and six DEmiRNAs were identified to be involved in the circRNA–miRNA–mRNA regulatory network. Notably, analysis of DEmRNA in the ceRNA network by cytohubba showed novel_circ_0002771/gga-let-7g-3p/*CDC20*, novel_circ_0000396/gga-miR-193a-3p/*CTNNB1* as the key regulatory process. Studies show that *CDC20* is the central gene of bone marrow mesenchymal stem cells for cartilage [37]. Wang et al. [38] found that CDC20 was expressed in cartilage progenitor cells through single-cell sequencing of healthy human chondrocytes. Liu et al. [37] showed that CDC20 may be the central gene for bone marrow mesenchymal stem cells to initiate osteogenesis, adipogenesis and cartilage. Secondly, knockout of CDC20 inhibits osteogenic differentiation of craniofacial mesenchymal stem cells [39]. CDC20 is significantly downregulated in the cartilage transcriptome of osteoarthritis mice [40]. These results suggest that CDC20, as a key gene in the cell cycle, plays an important role in the regulation of cartilage growth. Our tissue expression profiling results showed that the CDC20 gene in VVD broiler chickens was significantly reduced in cartilage, and we speculated that cartilage tissue growth and development were slow. CTNNB1 is a key gene in the wnt/β-catenin signaling pathway. Mice without the CTNNB1 gene showed inhibition of the pathway and inhibited expression of the PRG4 gene, disrupting articular cartilage homeostasis and reducing osteogenic differentiation [41,42]. Secondly, studies have shown that miR-330-3p can slow cartilage degeneration in mice with osteoarthritis by targeting CTNNB1 [43]. SOX9 binding competes with the TCF/LEF binding site within *CTNNB1*, thereby inhibiting Wnt signaling and participating in chondrocyte differentiation [44]. We speculated that two key ceRNA axes in our results were closely associated with chondrocyte proliferation and differentiation, but their exact mechanisms remain unknown.

## 5. Conclusions

In conclusion, this study provided the first comprehensive analysis of circRNAs, miRNAs and mRNAs in VVD and normal Hubbard broiler leg cartilage. ceRNA networks were subsequently constructed and key ceRNA regulatory axes were screened. Our results play an important role in unraveling the development of VVD and lay the foundation for understanding the molecular mechanisms underlying the development of the disease. However, unfortunately, we did not provide more transcriptome samples and more rigorous differential settings to further clarify the generalizability of the VVD broiler cartilage transcriptome results.

## Figures and Tables

**Figure 1 genes-14-00622-f001:**
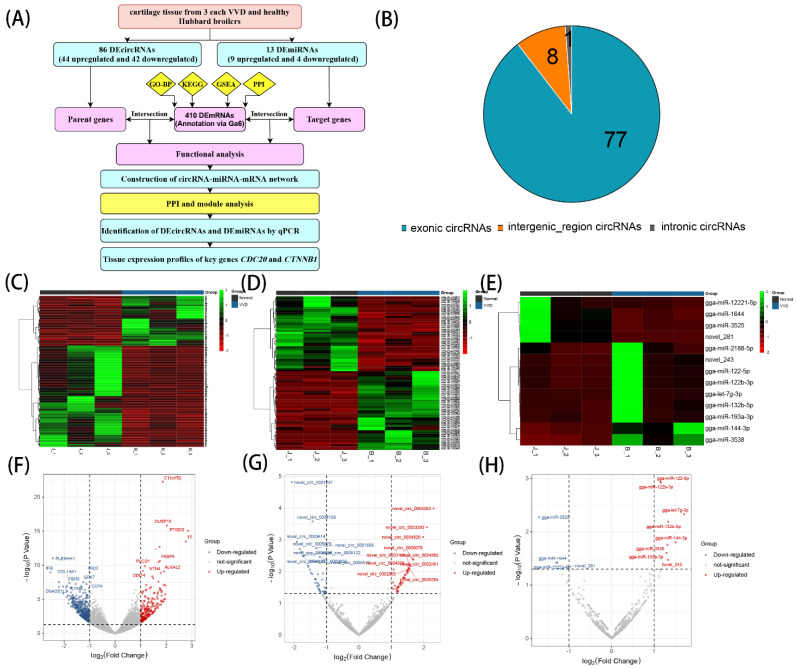
Flowchart of the research design and identification of differentially expressed circRNAs, miRNAs and mRNAs: (**A**) flowchart of the research design; (**B**) analysis of the type of DEcircRNAs; (**C**) heatmap of DEmRNAs; (**D**) heatmap of DEcircRNAs; (**E**) heatmap of DEmiRNAs; (**F**) volcano plot of DEmRNAs; (**G**) volcano plot of DEcircRNAs; (**H**) volcano plot of DEmiRNAs.

**Figure 2 genes-14-00622-f002:**
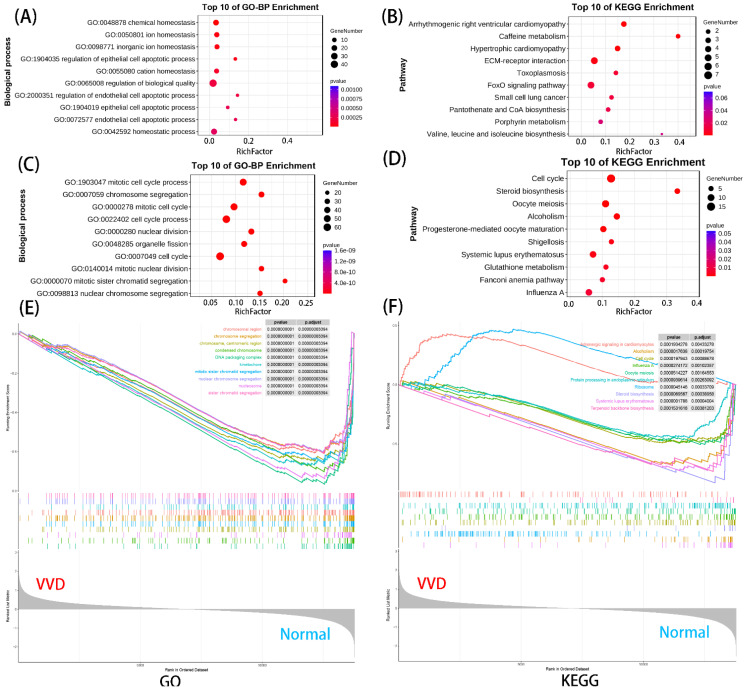
Functional enrichment analysis of the upregulated and downregulated DEmRNAs: (**A**,**B**) the top 10 GO-BP and KEGG enrichment analysis of upregulated DEmRNAs; (**C**,**D**) the top 10 GO-BP and KEGG enrichment analysis of downregulated DEmRNAs; (**E**,**F**) GSEA enrichment analysis of mRNAs.

**Figure 3 genes-14-00622-f003:**
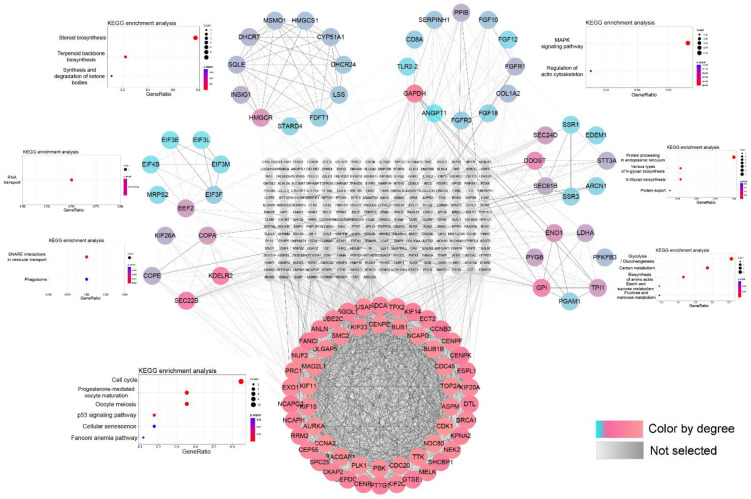
Seven modules extracted from Protein–Protein Interaction (PPI) network and KEGG enrichment analysis. The colors from pink to blue represent the degree of gene connectivity.

**Figure 4 genes-14-00622-f004:**
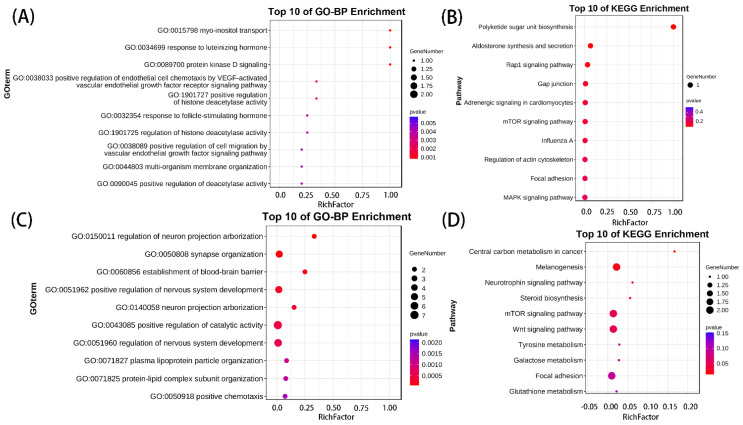
Functional enrichment analysis on DEcirc/miRNAs-DEmRNAs overlapping genes: (**A**,**B**) the top 10 GO-BP and KEGG enrichment analyses of DEcircRNAs–DEmRNAs overlapping genes; (**C**,**D**) the top 10 GO-BP and KEGG enrichment analyses of DEmiRNAs–DEmRNAs overlapping genes.

**Figure 5 genes-14-00622-f005:**
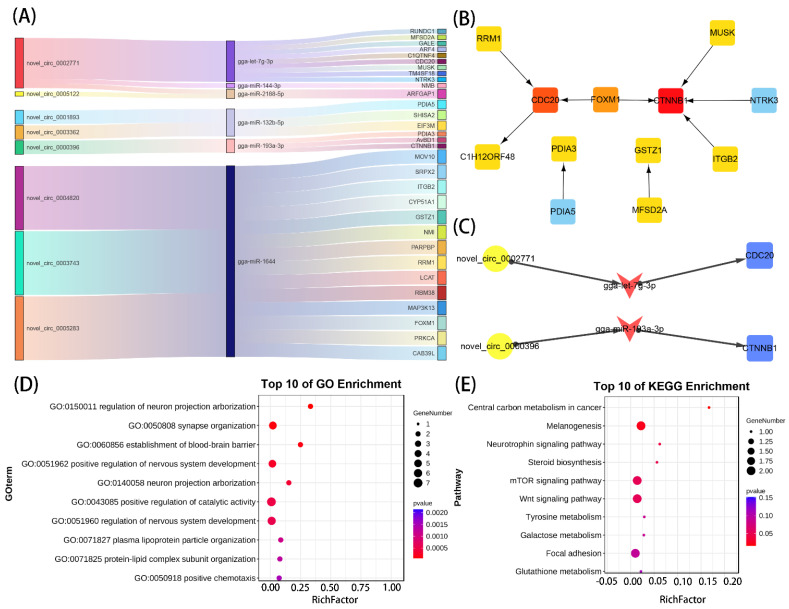
Construction of ceRNA network and screening of key regulatory axes: (**A**) Sankey diagram of ceRNA network; (**B**) PPI network of genes from the ceRNA network; (**C**) novel_circ_0002771/gga-let-7g-3p/CDC20, novel_circ_0000396/gga-miR-193a-3p/CTNNB1 regulatory axes; (**D**,**E**) top 10 GO-BP and KEGG enrichment analyses.

**Figure 6 genes-14-00622-f006:**
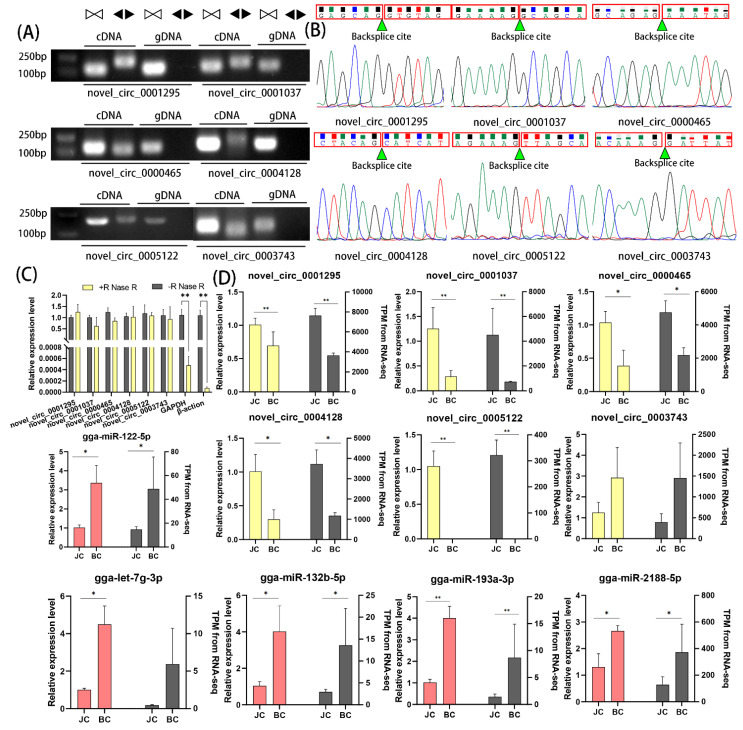
RT-qPCR validation of DEcircRNAs and DEmiRNAs: (**A**) Divergent primers amplified circRNAs in cDNA but not genomic DNA (gDNA). White triangles represent convergent primers and black triangles represent divergent primers; (**B**) Sanger sequencing confirmed the back-splicing junction sequence of circRNAs; (**C**) RT-qPCR showed resistance of circRNAs to RNaseR digestion. (**D**) RT-qPCR validation of six DEcircRNAs and five DEmiRNAs. Data were expressed as mean ± SEM. ** *p* < 0.01, * *p* < 0.05.

**Figure 7 genes-14-00622-f007:**
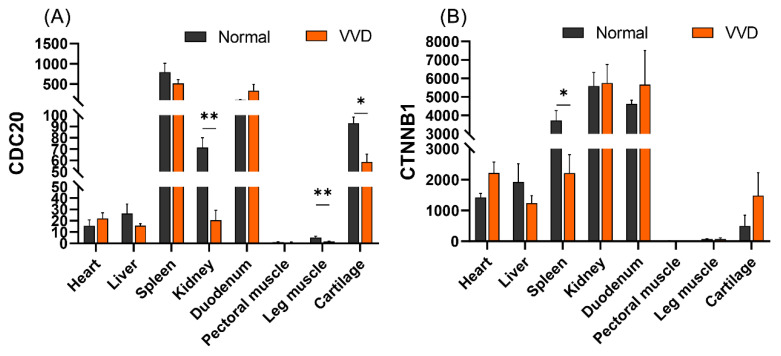
The expression profiles of CDC20 and CTNNB1 in VVD and normal broiler tissues: (**A**) tissue expression profiles of CDC20 in VVD and normal broilers; (**B**) tissue expression profiles of CTNNB1 in VVD and normal broilers. Data were expressed as mean ± SEM. ** *p* < 0.01, * *p* < 0.05.

**Table 1 genes-14-00622-t001:** Genes primers used in the real-time quantitative PCR.

Gene	Primer Sequence (5′-3′)	Amplification Length
novel_circ_0001295 Divergent primers	F: CCAGGCAATCGATGAGGAGT R: CATCCACGACGCTTGCAAAA	197 bp
novel_circ_0001295 Convergent primers	F: TACATCGCAGAAGTTGCTCCAC R: ATCCATCCTTCACGAGGTAGC	130 bp
novel_circ_0001037 Divergent primers	F: GCTCCAGAATGCATCAGAGGT R: TTCCGGTCTCCAATCACTGC	175 bp
novel_circ_0001037 Convergent primers	F: ACCGGAACTCTGGGAAAAGC R: TACCCCTTTCCATTCCTGCG	137 bp
novel_circ_0000465 Divergent primers	F: TACGTGGCTGGGCTTAAAGG R: TGGTCTGTAGGAGTCGATGC	130 bp
novel_circ_0000465 Convergent primers	F: AAGAAACAGTCCTAGCGAGCA R: ATCACCTCTTCCGTACTCCAA	146 bp
novel_circ_0004128 Divergent primers	F: TATGTCTTCGATGGTGCCTGT R: TTGCACTGTCATAGAGGGAGC	160 bp
novel_circ_0004128 Convergent primers	F: AAGCAGAGCGATCTTCCGAC R: CACAGGCACCATCGAAGACA	182 bp
novel_circ_0005122 Divergent primers	F: ATGTGATTGTGGATCCCGTCG R: TTCCCATCAACTGGTCTGCT	199 bp
novel_circ_0005122 Convergent primers	F: CTTGGAGTGTGCGGTGTCT R: CCACTCATAGCACGTTGGGT	190 bp
novel_circ_0003743 Divergent primers	F: GCCCGTACCAGACAAGGATA R: GTGGTAATCCTGTTGTGGGTCT	139 bp
novel_circ_0003743 Convergent primers	F: GCATAACGGAGAGCACAGTGA R: CACCTGTGTCTTTGTTGGTCAG	182 bp
gga-miR-122-5p	F: GCGTGGAGTGTGACAATGGT R: AGTGCAGGGTCCGAGGTATT	67 bp
gga-let-7g-3p	F: CGCTGTACAGGCCACTGC R: GCAGGGTCCGAGGTATTCGC	66 bp
gga-miR-132b-5p	F: CGCGACCATGGCTGTAGAC R: AGTGCAGGGTCCGAGGTATT	65 bp
gga-miR-193a-3p	F: CGCGAACTGGCCTACAAAGT R: AGTGCAGGGTCCGAGGTATT	66 bp
gga-miR-2188-5p	F: GGCGAAGGTCCAACCTCACA R: AGTGCAGGGTCCGAGGTATT	66 bp
CDC20	F: CTCTCCAGTGGGTCACGAAC	108 bp
	R: CTTGAGTCCGCACACCTCTT	
CTNNB1	F: TGCTGACTACCCAGTTGATGG	199 bp
	R: AGATACTAGCCCACCCCTCG	

## Data Availability

The data were submitted to the Genome Expression Omnibus (Accession Numbers PRJNA601858 and PRJNA910488) in NCBI.

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
