# Peer review of "Integration Analysis of circRNA–miRNA–mRNA and Identification of Critical Networks in Valgus-Varus Deformity (Gallus gallus)"

_genes, 2023, doi:10.3390/genes14030622_

Round 1

Reviewer 1 Report

This study was focused on Valgus-valgus deformity (VVD) which is a common leg deformity in broilers with inward or outward deviation of the tibiotarsus and tarsometatarsus. The authors wanted to study the roles of circRNA/miRNA/mRNA in VVD by using the competing endogenous RNA (ceRNA) networks. They analyzed the transcriptome data and showed the differentially expressed RNAs between the VVD groups and the control groups. These findings are interesting and meaningful. However, the English expression needs to be improved and here are some concerns.

Major concerns

1 Could the authors please explain why 35-day-old Hubbard broilers were chosen for sequencing and not other breeds and ages.

2 Readers who do not have expertise in the genetic techniques used in this study may find it difficult to understand some of the abbreviations, especially in the paper where they appear for the first time. I would suggest explaining what the abbreviations are, and putting then into the context.

3 The role of ceRNA in VVD and other skeletal diseases is not evident through the context of the article, and the authors should enrich the relevant background knowledge.

4 The authors eventually obtained two important targets, CDC20 and CTNNB1, and obtained tissue expression of both, but why were these two targets chosen in this study? Evidence of CDC20, CTNNB1 and skeletal relevance is not clearly indicated in the discussion. 

5 Why did the authors use GSEA for DEmRNA co-analysis? In line 171, more analysis will have more confusing results, and the authors are invited to summarize the main lines.

Minor concerns

1 In line 76, the author refers to the “same gender” in the experimental material, please specify this.

2 Authors should standardize italics and case in figure notes.

3 It is suggested to cite some references in recent years.

4 I don't quite understand what “DEmiRNA-DEmRNA and DEcircRNA-DEmRNA regulated relationship” means.

5 Image details need to be improved, such as figure 7.

6 Please provide GSEA R language code.

7 English language of this article needs to be improved.

Author Response

Dear reviewer,

Thank you very much for working our paper.

We have revised our manuscript according to the comments. These comments were valuable and helpful for revising and improving our paper as well as for providing important overall guidance for our study. We have carefully studied the comments and made corrections, which we hope will meet with your approval. In order to facilitate your examination, any changes we make to the original manuscript is highlighted in red in our revised manuscript.

Response to Review 1’s comment

Major concerns:

1 Could the authors please explain why 35-day-old Hubbard broilers were chosen for sequencing and not other breeds and ages.

Response 1: Thank you for your comments. VVD broiler chickens can be divided into three types according to their varus angle: mild deformity (0-15°, score = 1), intermediate deformity (15°-45°, score = 2) and severe deformity ( > 45°, score = 3). 35-day-old VVD broilers are in the state of severe deformity. They are selected for sequencing analysis, which provided more favorable evidence for subsequent studies.

2 Readers who do not have expertise in the genetic techniques used in this study may find it difficult to understand some of the abbreviations, especially in the paper where they appear for the first time. I would suggest explaining what the abbreviations are, and putting then into the context.

Response 2: Thank you for your comments. We have perfected the abbreviations in the article. The results are shown in lines 50, 52, 55, 65-67 of the article marked in red.

3 The role of ceRNA in VVD and other skeletal diseases is not evident through the context of the article, and the authors should enrich the relevant background knowledge.

Response 3: Thank you for your comments. CeRNA and other bone disease-related literature have been supplemented in lines 61-68 of the article’s introduction marked in red.

4 The authors eventually obtained two important targets, CDC20 and CTNNB1, and obtained tissue expression of both, but why were these two targets chosen in this study? Evidence of CDC20, CTNNB1 and skeletal relevance is not clearly indicated in the discussion.

Response 4: Thank you for your comments. We analyzed whole transcriptome sequencing through the synthesis of GO-BP, KEGG, GSEA, and PPI proteins, and obtained 2 key ceRNA regulatory axes, and identified two cell cycle-related genes, CDC20 and CTNNB1, which are associated with chondrocytes growth. The relevant literature has been supplemented in lines 309-323 of the article’s discussion marked in red.

5 Why did the authors use GSEA for DEmRNA co-analysis? In line 171, more analysis will have more confusing results, and the authors are invited to summarize the main lines.

Response 5: Thank you for your comments. GSEA analysis is an analysis method to sequence the differential expression degree of all genes produced by sequencing and enrich with a preset gene set, GSEA enrichment analysis can reduce the error caused by too few genes in traditional enrichment analysis, and can be combined with GO and KEGG. In addition, the main line is summarized in lines 181-183 of the article.

Minor concerns

1 In line 76, the author refers to the “same gender” in the experimental material, please specify this.

Response 1: Thank you for your comments. We selected male broilers that are more susceptible to disease as sequencing samples and marked them in line 86 of the article.

2 Authors should standardize italics and case in figure notes.

Response 1: Thank you for your comments. We have corrected the article for case and italics.

3 It is suggested to cite some references in recent years.

Response 3: Thank you for your comments. We have included additional literature citations in the introduction and discussion, which are highlighted in red.

4 I don't quite understand what “DEmiRNA-DEmRNA and DEcircRNA-DEmRNA regulated relationship” means.

Response 4: Thank you for your comments. We intersect the target gene of DEmiRNA and DEmRNA, and the host gene of DEcircRNA and DEmRNA for bioinformatics analysis.

5 Image details need to be improved, such as figure 7.

Response 5: Thank you for your comments. We have modified the details of figure 7.

6 Please provide GSEA R language code.

Response 6: Thank you for your comments. The GSEA code is as follows:

rm(list = ls())

if (!require("BiocManager", quietly = TRUE))

  install.packages("BiocManager")

BiocManager::install("clusterProfiler", force = TRUE)

BiocManager::install("enrichplot",force = TRUE)

install.packages("data.table")

library(clusterProfiler)

library(enrichplot)

library(ReactomePA)

library(data.table)

options(scipen = 200)

genelist_input <- read.table('mRNADE.txt',header = T,row.names = NULL)

geneList <- genelist_input[,c(1,3)]

head(geneList)

colnames(geneList) <- c('gene','logFC')

FCgenelist <- geneList$logFC #numeric vector

names(FCgenelist) <- as.character(geneList$gene) #named vector

FCgenelist <- sort(FCgenelist,decreasing=T) #decreasing order

head(FCgenelist)

7 English language of this article needs to be improved.

Response 7: Thank you for your comments. We have made moderate modifications to English grammar.

Reviewer 2 Report

In this manuscript reported the mRNA, circRNAs, and miRNAs transcriptomic differences between cartilage tissue of VVD and normal broilers using RNA-seq followed by functional analysis. The authors claimed that some biological processes of the ceRNA networks are an important role in skeletal development, potentially affecting the development of VVD in broilers. The manuscript in general is sound and well written. 

Major suggestion

1. The transcriptome analysis is based on a total of 6 samples (3 VVD and 3 normal), a too small number of samples to provide serious scientific speculations. Results are to some extent limited to a small number of replicates. The authors should mention the limitation of the sample size. In addition, the log2 FC > 1 is a very weak signal of biological evidence.

2. The method of Tissue expression profiles of CDC20 and CTNNB1 was not indicated in Materials and Methods

Minor suggestions

1. L33: Use the full name “competing endogenous RNA” for ceRNA in the Keywords.

2. L40: Since the author describes the VVD symptoms, the word “feeding and watering disorders” is not sound. Do the authors mean that chickens lose food and water intake?

3. L52: Use the full name “nucleotides” instead “nt”, which is the word that is mentioned only one time.

4. L56: Please indicate number 15 for the reference “Salmena et al.” in the text.

5. L61: Do they have any report in ceRNA and leg disease by the previous study? If yes, the author should add the reference here.

6. The objective of the study should be clearly stated in the last paragraph of the Introduction.

7. Figure 1. should move to the Result.

8. How about the RNA integrity number that was used in this study?

9. L269-270: Please give references for this sentence “VVD is a critical broiler leg disease with complex etiology, however, there are few studies on the molecular mechanism of VVD”.

10. L270-272: Use the capital “P” for the initial character of the sentence “previous studies have shown that heritability of VVD is 0.21-0.40 and bone quality ranged from 0.10 to 0.77 [26-28], and genetic selection can significantly reduce the incidence of VVD.”

Author Response

Dear reviewer,

Thank you very much for working our paper.

We have revised our manuscript according to the comments. These comments were valuable and helpful for revising and improving our paper as well as for providing important overall guidance for our study. We have carefully studied the comments and made corrections, which we hope will meet with your approval. In order to facilitate your examination, any changes we make to the original manuscript is highlighted in red in our revised manuscript.

Major concerns:

  1. The transcriptome analysis is based on a total of 6 samples (3 VVD and 3 normal), a too small number of samples to provide serious scientific speculations. Results are to some extent limited to a small number of replicates. The authors should mention the limitation of the sample size. In addition, the log2 FC > 1 is a very weak signal of biological evidence.

Response 1: Thank you for your comments. This study refers to a sample size of 3 biological replicates and |log2 FC| > 1 of a previous study of the broiler leg disease transcriptome, but we acknowledge the existence of the problem and we have added a portion of the conclusion in the article. The modifications are as follows:

“However, unfortunately, we did not provide more transcriptome samples and more rigorous differential settings to further clarify the generalizability of the VVD broiler cartilage transcriptome results.”

  1. The method of Tissue expression profiles of CDC20 and CTNNB1 was not indicated in Materials and Methods.

Response 2: Thank you for your comments. We have added tissue expression profiles of CDC20 and CTNNB1 to the materials and methods, which are shown in lines 160-167 of the article marked in red. The modifications are as follows:

“2.10. Expression profiles of CDC20 and CTNNB1

The total RNA from heart, liver, kidney, duodenum, pectoral muscle, leg muscle and cartilage was extracted using the traditional Trizol method and reverse tran-scribed into cDNA using the kit (Vazyme China). qPCR reaction conditions were set to 40 cycles of 95°C for 30s, 95°C for 10s, 60°C for 30s; 95°C for 15s, 60°C for 1 min and 95°C for 15s. The primer sequences are shown in Table 1. Three technical replicates were performed for each sample, and the expression levels of CDC20 and CTNNB1 in the tissues were calculated using the 2-method method.”

Minor concerns

  1. L33: Use the full name “competing endogenous RNA” for ceRNA in the Keywords.

Response 1: Thank you very much and we have changed it.

  1. L40: Since the author describes the VVD symptoms, the word “feeding and watering disorders” is not sound. Do the authors mean that chickens lose food and water intake?

Response 2: Thank you for your comments. We wanted to express that broilers have difficulty feeding and drinking due to lameness, and after our discussion, we deleted this word “feeding and watering disorders”.

  1. L52: Use the full name “nucleotides” instead “nt”, which is the word that is mentioned only one time.

Response 3: Thank you very much and we have changed it.

  1. L56: Please indicate number 15 for the reference “Salmena et al.” in the text.

Response 4: Thank you very much and we have indicated it in line 57 of the article.

  1. L61: Do they have any report in ceRNA and leg disease by the previous study? If yes, the author should add the reference here.

Response 5: Thanks for your reminder, we have added the relevant literature in lines 62-69 of the article marked in red.

  1. The objective of the study should be clearly stated in the last paragraph of the Introduction.

Response 6: Thanks to your suggestion, we have presented the purpose of the experiment more clearly in the introduction section, and the result of the revision is as follows:

“This study aims to screen the ceRNA regulatory axis and key genes associated with cartilage development through transcriptomic studies, laying the foundation for an in-depth investigation of the pathogenesis of VVD broiler.”

  1. Figure 1. should move to the Result.

Response 7: Thank you for your reminder, we have changed the location of figure 1.

  1. How about the RNA integrity number that was used in this study?

Response 8: The RNA integrity in this experiment was good, and the electrophoretic bands were clear and did not show any dragging.

  1. L269-270: Please give references for this sentence “VVD is a critical broiler leg disease with complex etiology, however, there are few studies on the molecular mechanism of VVD”.

Response 9: Thank you for your reminder, we have added relevant literature for this sentence.

  1. L270-272: Use the capital “P” for the initial character of the sentence “previous studies have shown that heritability of VVD is 0.21-0.40 and bone quality ranged from 0.10 to 0.77 [26-28], and genetic selection can significantly reduce the incidence of VVD.”

Response 10: Thank you for your reminder, we have corrected the uppercase and lowercase issue in the article.

Reviewer 3 Report

Well planned study which used powerful tools to answer research questions. However there are some concerns that I think the authors should address.

ceRNA is the abbreviation for what?

Introduce DE in line 64- differentially expressed

Figure 1a- don’t understand why DEcircDNA linked to parent genes, unlike DEmiRNAs linked to target genes. what is parent genes?

Line 77- same sex means male or female?

3 animals per group- too small

Didn’t justify the importance of studying DEcircRNA and miRNA- failed to highlight the importance of the study

Line 214-216- revise the sentence 

Line 270- Typesetting mistake, should capital P

Didn’t explain why study expression of CTNNB1 and CDC20 in other tissue and why were those tissues selected? 

What is the significance of lower expression of CTNNB1 in spleen in VVD? 

Discussion is ok even though not much compared results obtained with literature 

Author Response

Dear reviewer,

Thank you very much for working our paper.

We have revised our manuscript according to the comments. These comments were valuable and helpful for revising and improving our paper as well as for providing important overall guidance for our study. We have carefully studied the comments and made corrections, which we hope will meet with your approval. In order to facilitate your examination, any changes we make to the original manuscript is highlighted in red in our revised manuscript.

1 ceRNA is the abbreviation for what?

Response 1: Thank you for your comments. The full name of “ceRNA” is “competing endogenous RNA”.

2 Introduce DE in line 64- differentially expressed

Response 2: Thank you for your comments. We have added the DE annotation in line 73 marked in red.

3 Figure 1a- don’t understand why DEcircDNA linked to parent genes, unlike DEmiRNAs linked to target genes. what is parent genes?

Response 3: Thank you for your comments. Most of the circRNAs found so far come from exonic regions of genes, and the genes that produce circRNAs, we call host genes or parent genes. The introduction of parental genes provides a new mind for the analysis of DEcircRNA.

4 Line 77- same sex means male or female?

Response 4: Thank you for your comments. We chose male broilers as experimental subjects, and this has been corrected in the text.

5 3 animals per group- too small

Response 5: Thank you for your comments. This study refers to a sample size of 3 biological replicates of a previous study of the broiler leg disease transcriptome, but we acknowledge the existence of the problem and we have added a portion of the conclusion in the article. The modifications are as follows:

“However, unfortunately, we did not provide more transcriptome samples and more rigorous differential settings to further clarify the generalizability of the VVD broiler cartilage transcriptome results.”

6 Didn’t justify the importance of studying DEcircRNA and miRNA- failed to highlight the importance of the study

Response 6: Thank you for your comments. In this study, we screened the regulatory axis of circRNA/miRNA/mRNA and key genes by DEcircRNA, miRNA and mRNA, and we emphasize the role of the latter for the article. Therefore, we did not elaborate too much on DEcircRNA and miRNA.

7 Line 214-216- revise the sentence

Response 7: Thank you for your comments. We have refined and modified the statements. The modifications are as follows:

“among which, 31 mRNAs were overlapped with the DEmRNAs of annotating via Ga6 versions. Eventually, the Sankey diagram was used to show the correspondence of the three (Figure 5A, B).”

8 Line 270- Typesetting mistake, should capital P

Response 8: Thank you for your reminder, we have corrected the uppercase and lowercase issue in the article.

9 Didn’t explain why study expression of CTNNB1 and CDC20 in other tissue and why were those tissues selected?

Response 9: Thank you for your comments. We further explain the reasons for probing the tissue expression of the two key genes and highlight them in red on lines 277-279 in the article. heart, liver, spleen, kidney, duodenum, pectoral muscle, hamstrings and cartilage are very important organism organ that governs the tissue metabolism, cell growth and development and cellular communication of the organism, which lays the foundation for our subsequent experiments.

The article was modified as follows:

“CDC20 is a key gene in the cell cycle and CTNNB1 is an adhesion-related gene in the wnt/β--catenin signaling pathway, both of which play important roles in cell growth and cartilage development.”

10 What is the significance of lower expression of CTNNB1 in spleen in VVD?

Response 10: Thank you for your comments. Studies have shown that the wnt/β-catenin signaling pathway CTNNB1 is associated with bone development, and in mice with CTNNB1 caused by osteoporosis caused by modification of osteocytes using constitutively active β-catenin allele in osteoblasts, it was found that the spleen contains dense and diffuse infiltrated neutrophils [1]. Secondly, a decrease in red blood cells is observed in the spleen [2]. Inhibition of CTNNB1 in the wnt/β-catenin pathway showed impaired terminal erythrocyte differentiation/reticulocyte maturation, resulting in fatal anemia [3]. Therefore, the CTNNB1 gene was significantly reduced in the spleen tissue of VVD broilers, which may indicate that VVD broilers are anemic and have an inflammatory response in the organism. This corresponds to our previous results [4].

[1] Kode, A., Manavalan, J., Mosialou, I. et al. Leukaemogenesis induced by an activating β-catenin mutation in osteoblasts. Nature 506, 240–244 (2014). https://doi.org/10.1038/nature12883.

[2] Kode, A., Mosialou, I., Manavalan, S. J. et al. FoxO1-dependent induction of acute myeloid leukemia by osteoblasts in mice. Leukemia, 30(1), 1–13 (2016). https://doi.org/10.1038/leu.2015.161.

[3] Heil, J., Olsavszky, V., Busch, K. et al. Bone marrow sinusoidal endothelium controls terminal erythroid differentiation and reticulocyte maturation. Nat Commun 12, 6963 (2021). https://doi.org/10.1038/s41467-021-27161-3.

[4] Li, J., Ma, Y., Zhang, L. et al. Valgus-varus deformity induced abnormal tissue metabolism, inflammatory damage and apoptosis in broilers. British poultry science, 1–10 (2022). Advance online publication. https://doi.org/10.1080/00071668.2022.2121640.

11 Discussion is ok even though not much compared results obtained with literature

Response 11: Thank you for your comments.